# Research on Modeling of a Micro Variable-Pitch Turboprop Engine Based on Rig Test Data

**Xiaochun Zhao** [1] , **Xianghua Huang** [1],* **and Tianqian Xia** [2]

1   JiangSu Province Key Laboratory of Aerospace Power System, Nanjing University of Aeronautics and Astronautics, NO.29 Yudao Street, Nanjing 210016, China; xiaochun1992@126.com
2   AECC Aero Engine Control System Institute, Binhu District, Wuxi 214013, China; xiatianqian123@163.com
*   Correspondence: xhhuang@nuaa.edu.cn; Tel.: +86-1385-181-7882

**Abstract:** Exact component characteristics are required for establishing an accurate component level aeroengine model. When component characteristics is lacking, the dynamic coefficient method based on test data, is suitable for establishing a single-input and single-output aeroengine model. When it is applied to build multiple-input, multiple-output aeroengine models, some parameters are assumed to be unchanged, which causes large error. An improved modeling method based on rig data is proposed to establish a double-input, double-output model for a micro variable-pitch turboprop engine. The input variables are fuel flow and pitch angle, and the output variables are rotational speeds of the core engine and the propeller. First, in order to gather modeling data, a test bench is designed and rig tests are carried out. Then, two conclusions are obtained by analyzing the rig data, based on which, the power turbine output is taken as the function of the core speed and the propeller speed. The established model has the property that the input variables can vary arbitrarily within the defined domain, without any restriction to the output variables. Simulation results showed that the model has a high dynamic and steady-state accuracy. The maximum error was less than 8%. The real-time performance was greatly improved, compared to the component level model.

**Keywords:** turboprop engine; model based on data; dynamic coefficient method

## 1. Introduction

Micro gas turbine engine (MGTE) is a gas turbine engine with thrust less than 100 daN. It has the advantages of low cost, high performance, small size, and easy maintenance and storage [1,2]. It can be used as the alternative power plants of micro air vehicles, unmanned combat air vehicle (UCAV), distributed generation applications, reconnaissance plane, and other small weapons [3,4]. A variety of micro turbojet engines have been developed. Rig tests of conventional aeroengine can only be carried out in a few research institutes, while the rig test of micro-engines can be carried out in the laboratory, with a low risk and cost; therefore, micro-engine rig tests can be used for verifying advanced control theory of aeroengines [5–7]. Advanced turboprop engines have two independent controlled variables, fuel flow, and propeller pitch angle, which makes it possible to keep the propeller rotating stably at variable speeds. Therefore, the turboprop engine can generate more power with fewer fuel consumption [7].

The control system of micro turbojet engine is a single-input and single-output control system, in which only the fuel flow is adjustable under normal working conditions. The micro turboprop test system with adjustable fuel flow and adjustable pitch angle is an ideal test platform for studying aeroengine modeling, multivariable control, and advanced control methods. The propeller pitch control mechanism of the conventional-size turboprop engine is complicated and costly, and its size is too large to be installed on a micro-engine. Therefore, micro turboprop engines that are commercially

available have fixed pitches. In this paper, a pitch regulating mechanism is designed and a micro turboprop engine test platform is built.

The mathematical model of the aeroengine plays an important role in the design and verification of the aeroengine control system, which can effectively reduce the development time, risk, and cost [8,9]. Micro gas turbine engines are widely used as low-cost solutions, therefore, fewer sensors are usually available than in standard gas turbines [8]. An accurate model can offer reference signals to the control system of micro engines. The modern aeroengine control systems adopt model-based ones, to fully utilize their performance. Modern aeroengine control methods such as performance seeking control, life extending control, and fault-tolerant control require an on-board engine model to track unmeasured parameters [10,11]. Aeroengine modeling methods can be divided into three categories—analytical methods based on operating principles (white box method), system identification methods based on test data (black box method), and gray box methods, which consider synthetic test data and operating principles.

A component level model is built based on analytical methods, by using component characteristics and engine operating principles. It is the most common aeroengine modeling method. Fitzgerald et al. from Georgia Tech Institute built a component level model, based on the component characteristics of SPT5 [12,13]. The component characteristics are obtained via rig tests carried out in their own test platform. The model obtained by the analytical method has a high precision and can simulate both steady-state and dynamic engine outputs in the full envelope. However, this method requires accurate component characteristic data and has heavy calculation burden. When the exact engine component characteristics cannot be obtained, the scaled general component characteristics are often used instead, which reduces the accuracy of the model [14,15]. The component level model needs to solve many equilibrium equations and iterations, which exceeds the ability of the embedded control unit. Aeroengine models with more computational efficiency are required for engineering application.

The dynamic coefficient method is a commonly used gray box modeling method. An accurate simplified model can be established by this method only with the engine test data. It can be used for the early-stage design and experimentation of the engine control system. Xin et al. established the mathematical model of a micro turbojet engine by using the dynamic coefficient method [16]. The micro turbojet engine is witnessed as a single-input and single-output control system, in which only fuel flow is adjustable under normal working conditions. Wenxiang et al. established a real-time model of a two-spool turbofan engine using the same method, with fuel flow as the only input [17]. The dynamic coefficient method used in the above two models was only applicable to engines with one input variable, which is commonly fuel flow. Dong et al. proposed an improved dynamic coefficient method and applied it to a two-spool turboshaft engine model with two inputs, fuel flow, and pitch angle [18]. However, this model is accomplished by regarding the propeller speed as constant and it is only suitable for engines with a closed-loop control. A variable-pitch turboprop engine model has two inputs, fuel flow, and pitch angle. Parameters such as propeller speed, core speed, compressor exit total pressure, and turbine exit total temperature change as the inputs change. Therefore, the dynamic coefficient method mentioned above cannot be applied to this model.

Based on the analysis of the working principle and the test data of the micro turboprop engine, a new modeling method based on test data is proposed by improving the traditional dynamic coefficient modeling method. The power output of the core engine is designed as the function of two speeds—the core speed and the propeller speed. The dynamic coefficient model of the core engine and the propeller power model are, respectively, established. Then, the model of micro variable-pitch turboprop engine that integrates the two models is accomplished.

The article is organized as follows. Section 2 describes the overall design and the structure of the model. Section 3 describes the design of the rig test bench and the test program. Section 4 is about the processing and analysis of the test data. Section 5 describes the details of the model, respectively, the detailed model structure, the steady state model, and the dynamic model of the core engine and the propeller. Section 6 is about the results of the model simulations.

## 2. Overall Design of the Model

The engine is a two-spool turboprop engine with free turbine. The core engine consists of a centrifugal compressor, a combustor, and a high-pressure turbine. A power turbine is installed behind the core engine and converts the heat of high-temperature gas from the core engine into power. The propeller is connected to the power turbine through a reducer.

The overall structure of the micro turboprop model is shown in Figure 1. The model has two inputs, fuel flow $W_f$ and pitch angle $\beta$. The outputs are the core speed $N_h$ and the propeller speed $N_p$. The model integrates a core engine model with a propeller model. The propeller model is used to calculate the propeller power absorbed from the core engine. Due to the lack of exact engine component characteristics, a component level model with satisfying precision could not be obtained. In the traditional dynamic coefficient method, the steady state relationship between speed and fuel flow and the acceleration coefficient were obtained by interpolation. It is commonly used for the modeling of single-input, single-output system and cannot be directly used for the modeling of variable pitch turboprop engine. Therefore, an improved dynamic coefficient method based on the rig test data is proposed to solve this problem.

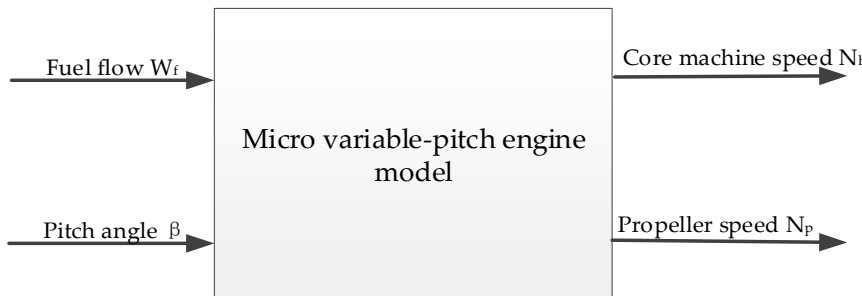

**Figure 1.** Overall structure of the model.

## 3. Design of the Rig Test Bench and Test Program

The SPT10 micro engine made by the JetPT company, is a two-spool turboprop engine with a free turbine. It can work in a turboshaft or turboprop operating mode, depending on the load of the output shaft. Since this engine has no variable-pitch propeller, a pitch regulating mechanism was designed. The schematic diagram and photo of the test bench are shown in Figure 2. The engine was fixed to the mounting frame, and the output shaft and the propeller were connected through a flexible coupling with torque sensor. The screw rod slider was driven by a stepping motor, to move axially, in order to realize the pitch adjustment of the propeller.

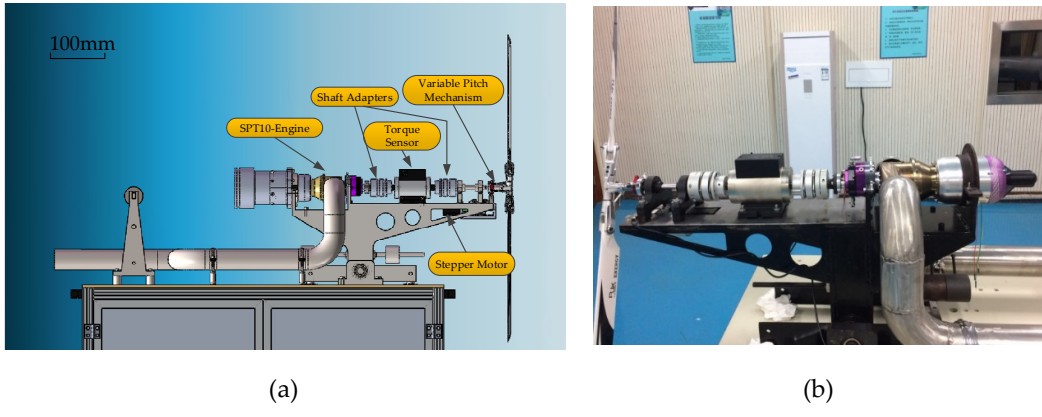

(a)          (b)

**Figure 2.** Test bench of the micro variable-pitch turboprop engine. (**a**) Schematic diagram; and (**b**) physical photo.

Various sets of pitch angle $\beta$ and fuel flow $w_f$ were input to the engine test system during the experiment. The measurements included the signals of the propeller speed $N_p$, the core speed $N_h$, and the output shaft torque $M$. As the fuel flow of a micro gas turbine was too low to find a flow meter with enough accuracy, a fuel pump speed was used to indirectly meter the fuel flow.

The designed test program consisted of two parts:

- The first part of the test consisted of 2 stages. For the convenience of expression, they were expressed as stage $N_{hv1}$ and stage $N_{hv2}$. In this part, $\beta$ remained unchanged, and $N_h$ was adjusted by changing $W_f$. In order to verify the reproducibility of the model, two tests were performed at this mode, called $N_{hv1}$ and $N_{hv2}$, respectively. The change rates of the fuel flow were different in the two stages. First, the power lever angle was changed to the maximum state within 2 to 5 s from the idle state. For safety, the maximum test state was defined as the state with a core speed of 56,000 r/min, and the propeller speed was 1500 r/min in the maximum state. After the engine speed was stabilized, the throttle was moved back to the idle state.
- The other part of the test consisted of 4 stages, expressed as stage $N_{h1}$, $N_{h2}$, $N_{h3}$, and $N_{h4}$. In this part, in order to obtain the characteristic parameters of the variable pitches, tests were carried out at 4 core speeds. The selected speeds were $N_{h1} = 35709$ r/min, $N_{h2} = 39571$ r/min, $N_{h3} = 44981$ r/min, and $N_{h4} = 50386$ r/min, and they were higher than that in an idle state. $\beta$ was set at one of the 5 selected degrees at each operating node, they were $\beta_1 = 21.875°$, $\beta_2 = 27.3438°, \beta_3 = 32.8125°$, $\beta_4 = 38.2813°$, and $\beta_5 = 43.75°$. When carrying out the experiments, $W_f$ remained unchanged after $N_h$ was maintained at 4 selected speed, respectively. $\beta$ changed between the 5 selected degrees in turn. $\beta$ did not switch to the next degree until $N_p$ stayed stable for some time.

## 4. Processing and Analysis of Test Data

The raw data obtained from the test showed large fluctuations and could not be directly used for modeling. Therefore, the test data need to be preprocessed first. The fluctuations were mainly high-frequency periodic noise. The fluctuations mostly likely resulted from the characteristics of sensors and supply voltage, because the frequency and amplitude almost remained the same when the input signals changed. Therefore, the original signal was preprocessed by the mean filter, thus, the average value of several continuous data were used. The test data of $N_p$, $N_h$, and $M$ showed less fluctuation and were processed with a mean filter at a depth of 20. The fuel supply $W_f$ was calculated with the fuel pump supply voltage. Due to the performance of the voltage sensor and the influence of the power ripple, fluctuations occurred in the measured voltage signal. When the fuel supply was small, fluctuations were relatively large, as in stage $N_{h1}$, shown in Figure 3d. The data of $W_f$ was processed by a mean filter at a depth of 50.

Power absorbed by propeller $P_v$ was calculated by

$$P_v = C_p\left(J_p, \beta\right)\rho_0\left(\frac{N_p}{60}\right)^3\left(2R_p\right)^5 \tag{1}$$

where $J_p = \frac{V_0}{2N_pR_p}$ is the propeller advance ratio, $V_0$ is the axial velocity of propeller, $R_p$ is the radius of propeller, $\rho_0$ is air density, and $C_p$ is power coefficient of propeller. $C_p$ is a function of $J_P$ and $\beta$. This equation was obtained on the basis of the integral of the absorbed power times blade foil, and was not affected by the working condition of the propeller.

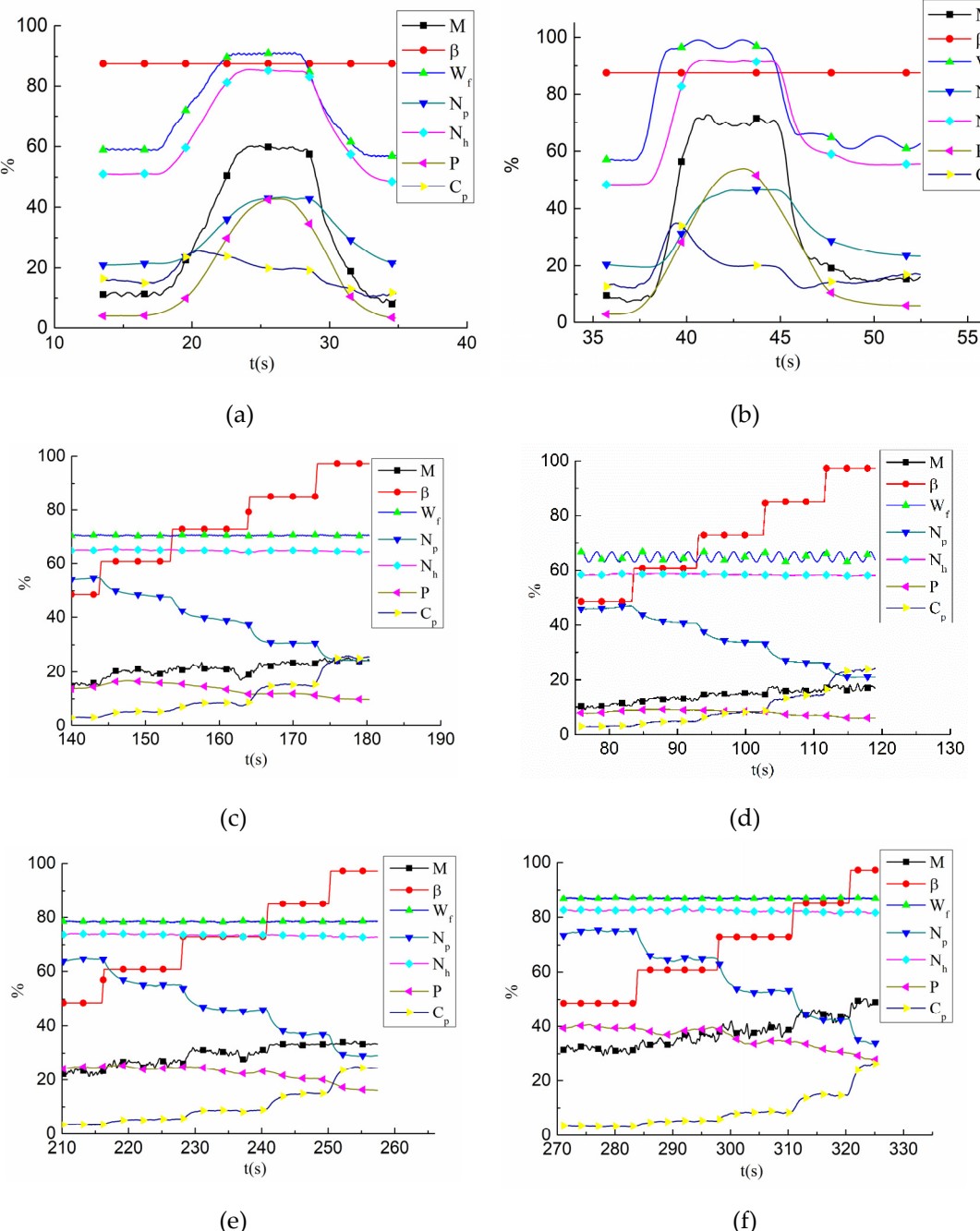

**Figure 3.** Test results of 6 stages. (**a**) Changing $W_f$ while keeping $\beta$ unchanged at stage $N_{hv1}$; (**b**) changing $W_f$ while keeping $\beta$ unchanged at stage $N_{hv2}$; (**c**) changing $\beta$ while keeping $N_h$ unchanged at stage $N_{h1}$; (**d**) changing $\beta$ while keeping $N_h$ unchanged at stage $N_{h2}$; (**e**) changing $\beta$ while keeping $N_h$ unchanged at stage $N_{h3}$; and (**f**) changing $\beta$ while keeping $N_h$ unchanged at stage $N_{h4}$.

During the rig test, $V_0 = 0$, so $J_p = 0$. Thus $C_p$ could be regarded as a monotropic function of $\beta$. $\rho_0$ and $R_p$ are constant. Equation (1) could be simplified as Equation (2), after merging the constants.

$$P_v = C'_p(\beta)N_p^3 \tag{2}$$

In order to facilitate the expression, the merged power coefficient $C'_p$ was simplified as $C_p$ in the following text.

$P_v$ in the rig test could be computed from the power turbine output torque $M$ and the propeller speed $N_p$:

$$P_v = \frac{2\pi N_p M}{60} \tag{3}$$

According to Equation (2), $C_p(\beta)$ could be obtained as:

$$C_p(\beta) = \frac{P_v}{N_p^3} \tag{4}$$

The calculated $C_p$ and $P_v$ were then normalized. The data processing results of the 6 test stages are shown in Figure 3. The sampling frequency was 50 Hz, and the sampling number in each stage was above 2500. The calculated $C_p(\beta)$ in different stages are shown in Table 1. In stage $N_{hv1}$ and $N_{hv2}$, the calculated power coefficient $C_p$ changed, while $\beta$ kept stable. It is because the power absorbed by propeller did not equal that of the turbine output in the acceleration and deceleration process, and the measured torque of the output shaft could not truly reflect the absorbed power. Two conclusions could be obtained from the test data:

1. When $W_f$ remained unchanged, $N_h$ barely changed with the variation of $\beta$. As shown in Figure 3c–f, the test results of $N_h$ basically remained the same, when only $\beta$ was changed.
2. $C_p$ was merely correlated with $\beta$. Neither $N_h$ nor $N_p$ had any influence on it. As shown in Table 1, when $\beta$ remained unchanged, $C_p$ did not vary with $N_h$. As shown in Figure 3c,d, $N_h$ and $N_p$ in test stage $N_{h1}$ were different from those in test stage $N_{h2}$ as $W_f$ in the two stages were different. However, there were no significant differences in $C_p$ in the two stages when $\beta$ was set to the same value.

These conclusions were the theoretical basis for the core engine dynamic coefficient model and the propeller speed dynamic mode.

**Table 1.** $C_p$ in different test stages.

| Stage | $C_{p1}$ | $C_{p2}$ | $C_{p3}$ | $C_{p4}$ | $C_{p5}$ |
|---|---|---|---|---|---|
| $N_{h1}$ | 3.08 | 4.62 | 8.06 | 14.00 | 23.96 |
| $N_{h2}$ | 3.21 | 5.15 | 8.41 | 15.31 | 24.97 |
| $N_{h3}$ | 3.45 | 5.00 | 8.782 | 14.74 | 24.51 |
| $N_{h4}$ | 3.45 | 4.87 | 8.44 | 15.01 | 25.73 |

$C_{p1}, C_{p2}, C_{p3}, C_{p4}$, and $C_{p5}$ represent $C_p$ when $\beta$ is $\beta_1 = 21.875°$, $\beta_2 = 27.3438°$, $\beta_3 = 32.8125°$, $\beta_4 = 38.2813°$, and $\beta_5 = 43.75°$, respectively.

## 5. Modeling Based on Rig Data by the Improved Dynamic Coefficient Method

### 5.1. Model Structure of the Micro Variable-Pitch Turboprop Engine

Figure 4 shows the model structure of the micro variable-pitch turboprop engine. The inputs were fuel flow $W_f$ and pitch angle $\beta$, and the outputs were core speed $N_h$, propeller speed $N_p$, engine output power $P_e$, and propeller absorbed power $P_v$. $P_v$ was calculated through the current $N_p$ and the power coefficient $C_p(\beta)$, which could be obtained by interpolation, according to Table 1. $N_h$ was obtained from the dynamic coefficient model of the core engine. $P_e$ was obtained by the two-dimensional interpolation of $N_h$ and $N_p$. In the power turbine model, $N_p$ was calculated according to the difference between $P_e$ and $P_v$. The propeller power coefficient interpolation table, the engine power two-dimensional interpolation table, and the core engine dynamic coefficient table were obtained from the test data.

The most important components of the model were the core engine dynamic coefficient model and the propeller speed dynamic model. According to Conclusion (1) in Section 4, $N_h$ could be regarded as a monotropic function of $W_f$ and the influence of $\beta$ could be neglected. Therefore, the core engine

model could be established by the dynamic coefficient method with $W_f$ as its input. According to Conclusion (2), the power coefficient of the propeller c

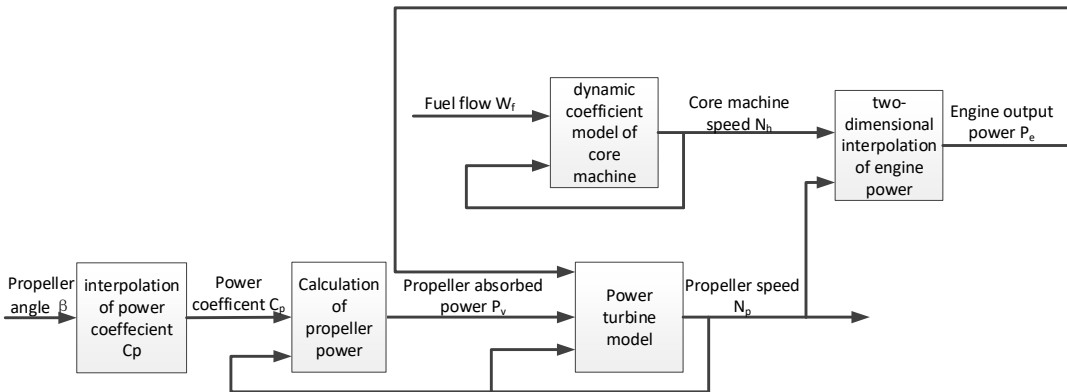

**Figure 4.** Model structure of micro variable-pitch turboprop engine.

### 5.2. Steady State Model of the Core Engine

The working state of power turbine had little effect on the core engine [19]. The rig data also indicated that the change of *β* alone could only lead to the change of the power turbine speed, which was proportional to $N_P$, but could not change the core speed $N_h$. Therefore, $N_h$ was the monotropic function of fuel flow $W_f$ in the steady state, that is:

$$N_{hs} = f\left(W_{fs}\right) \tag{5}$$

The subscript s represents the steady state. The function was implemented by one-dimensional interpolation.

The steady-state correlation correspondence between $N_h$ and $W_f$ was usually plotted as a "baseline". For the engine studied in the paper, the baseline and the actual operating line of the test stage $N_{hv1}$ was drawn in the same chart, as shown in Figure 5. During the acceleration process, the actual fuel consumption was larger than that in the steady state, with the same $N_h$. As a result, the acceleration line was below the baseline of process 1 shown in Figure 5. During deceleration, the actual consumed fuel was smaller than that in the steady speed, so the deceleration line was above the baseline process 2 shown in Figure 5.

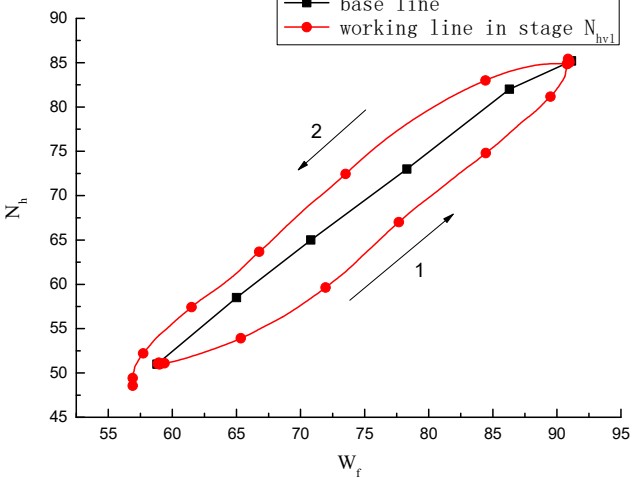

**Figure 5.** Baseline and actual working line in the test stage $N_{hv1}$.

### 5.3. Dynamic Model of the Core Engine

The acceleration or deceleration of the vehicle was achieved by changing the thrust produced by the propeller. The most common control method to change the thrust was to adjust the pitch angle $\beta$ of propeller, while keeping its speed constant. When $\beta$ changed, the propeller absorbed the power changes. In order to keep the propeller speed constant, the engine output power needed to be changed accordingly. The core speed was one of the decisive factors of the engine output power. It was also the basis for establishing the propeller speed model. Each speed $N_{hs}$ corresponded to a specific fuel flow $W_{fs}$ in the steady state. Excess fuel flow compared with $W_{fs}$ would result in an imbalanced torque, then the engine would enter into a dynamic process. According to Equations (3)–(5), the fuel flow required for a stable core speed is:

$$W_{fs} = f^{-1}(N_{hs}) \tag{6}$$

The excess fuel flow $\Delta W_f$ was defined as:

$$\Delta W_f = W_f - W_{fs} \tag{7}$$

If $\Delta W_f > 0$, it indicated that the fuel flow was larger than the required steady fuel flow and the power generated by the high pressure turbine was greater than the compressor consumed. Therefore, the engine was in the acceleration process. If $\Delta W_f < 0$, it indicated that the fuel supply was less than the required steady fuel supply and the high pressure turbine generated less power than the compressor consumed, then the engine was in the deceleration process. Considering that the accelerated speed was proportional to $\Delta W_f$ at certain speeds, the dynamic coefficient of the core engine at time $i$ could be defined as:

$$K_{Nh}(i) = \frac{N_h(i+1) - N_h(i)}{\Delta W_f(i)\Delta t} \tag{8}$$

where $\Delta t$ is the simulation step size.

Because of the strong nonlinearity of aeroengine, the dynamic coefficient $K_{Nh}$ differs at different core speed. Therefore, it is designed as the function of $N_h$:

$$K_{nh} = \varphi(N_h) \tag{9}$$

The dynamic performance of the engine in the acceleration process differed from that in the deceleration process, so the dynamic acceleration coefficient and the dynamic deceleration coefficient needed to be calculated separately.

According to the definitions above, the core speed could be calculated as:

$$N_h(t + \Delta t) = N_h(t) + K_{Nh} \times \Delta W_f(t) \times \Delta t \tag{10}$$

$$W_{fs}(t) = f^{-1}(N_{hs}(t)) \tag{11}$$

$$\Delta W_f(t) = W_f(t) - W_{fs}(t) \tag{12}$$

If the initial core speed and fuel supply law were known, the core speed at each moment could be calculated according to the above formulas.

### 5.4. The Dynamic Model of the Propeller Speed

When the engine was under closed-loop control, the propeller speed $N_p$ remained constant. The power turbine output power $P_e$ in the steady state could be approximately regarded to be the monotropic function of $N_h$ [16]:

$$P_e = \psi(N_h) \tag{13}$$

Based on this assumption, $P_e$ could be calculated according to the core speed $N_h$. However, $N_p$ was a variable under open-loop control. The engine output power might be different at the same $N_h$ because the working state of the power turbine changed with $N_p$. Due to this problem, Equation (13) could not be used to establish the model of the turboprop engine in open-loop state with two independent variables. The influences of $N_h$ and $N_p$ on engine output power were both considered and $P_e$ was designed as the function of $N_h$ and $N_p$:

$$P_e = \psi(N_h, N_P) \tag{14}$$

$P_e$ was calculated using a two-dimensional interpolation table whose data were acquired from the rig test.

For the rotating parts of turboprop engine, the rotor acceleration could be calculated as:

$$\dot{N}'_p(t) = \frac{M}{J} \tag{15}$$

where $J$ represents the inertia moment of the engine output shaft and the connecting shaft between engine and propeller. $M$ represents the torque:

$$M = \frac{60(P_e(t) - P_v(t))}{2\pi N'_p(t)} \tag{16}$$

By combining Equations (15) and (16), the rotor acceleration $\dot{N}_p$ was decided by the excess power, which was the difference between $P_e$ and the propeller consumed power $P_v$:

$$\dot{N}'_p(t) = \frac{30(P_e(t) - P_v(t))}{\pi \cdot J \cdot N'_p(t)} \tag{17}$$

$$N_p(t + \Delta t) = N_p(t) + \dot{N}_p \cdot \Delta t \tag{18}$$

where $J$ represents the inertia moment of the engine output shaft and the connecting shaft between the engine and the propeller, and $N'_p$ represents the core speed in unit of r/min.

## 6. Model Simulation Results

The improved dynamic coefficient method was used to establish the model of a micro variable-pitch turboprop engine. Simulations were carried out with the same inputs of the rig tests in 6 designed stages. The most noteworthy parameters were $N_p$ and $N_h$. As shown in Figure 6, the simulations results were compared with the rig data. The displayed data were $W_f$, $\beta$, $N_p$, and $N_h$ of both the simulation and the experiments.

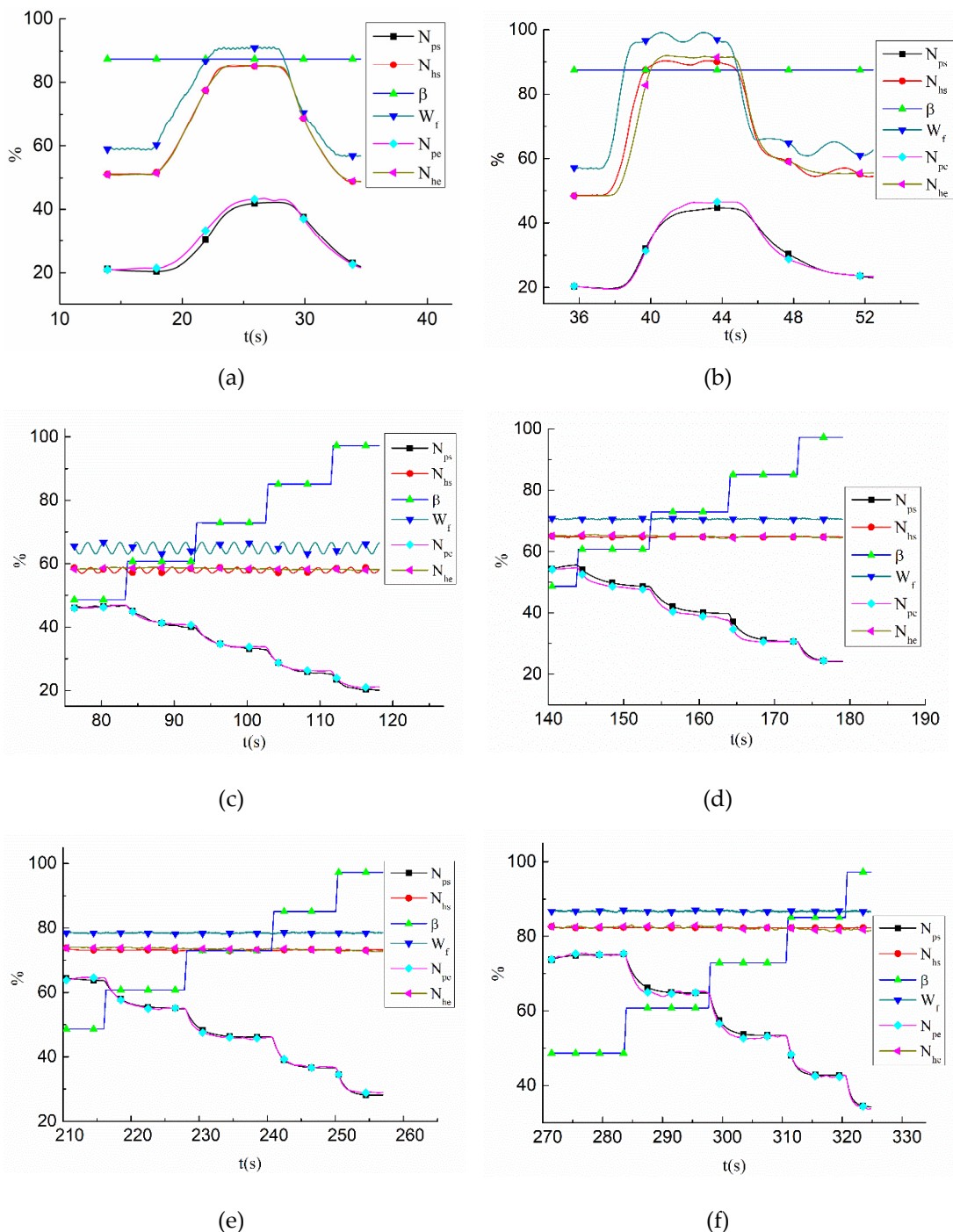

**Figure 6.** Simulation results compared with the rig data. The subscript 's' represents simulation and 'e' represents experiments. (**a**) Stage $N_{hv1}$; (**b**) stage $N_{hv2}$; (**c**) stage $N_{h1}$; (**d**) stage $N_{h2}$; (**e**) stage $N_{h3}$; (**f**) stage $N_{h4}$.

## 7. Discussion

The fitting effect of the model was characterized by the fitting error, which represent the largest difference between simulation and experiment, at the same time, in the full range. Since the core speed dynamic coefficient was obtained based on the rig data of the test stage $N_{hv1}$, the fitting effect was significantly good at this stage. In stage $N_{hv2}$, due to the acceleration rate and the deceleration rates being larger than those in stage $N_{hv1}$, the fitting error was bigger but was within an acceptable range. The simulation results were in good agreement with the test data in different acceleration

processes, which meant that the model based on the limited test data could be used in different working conditions. In the whole 6 test stages, the outputs of the model could track the test data well. The error in the whole process was less than 8%. The simulation results showed that the improved method had satisfied the modeling accuracy and could be applied to modeling the turboprop engine in all kinds of operating state with different core speeds and propeller speeds.

## 8. Conclusions

A rig test bench for the micro variable-pitch turboprop engine was designed. Rig test carried out on the designed test bench could provide satisfactory data for engine modeling.

Data reflecting the characteristics of the engine and the propeller were obtained by adopting a six-stage test program. Two conclusions of great importance to modeling were obtained by analyzing the data.

The model of a micro variable-pitch turboprop engine was established, based on the limited rig test data through the improved dynamic coefficient method. The improved method fixed the problem where traditional coefficient methods could not be applied for establishing a double-input, double-output model for turboprop engines. The model was established without any component characteristics of the engine, which provided a more easily realized method than the traditional component level model. The biggest difference between simulation and experiment in the whole process was less than 8%, which proved that the improved method had satisfied modeling accuracy. Compared with the component level model, this model avoided solving equilibrium equations and iterations and had better real-time performance.

Our future research will focus on improving the accuracy and efficiency, and the application of the model. More specifically, experiments with larger variations in $\beta$ and $w_f$ will be carried out and advanced methods for solving differential equations will be adopted. A control system run in an embedded controller, based on this model will be developed.

**Author Contributions:** Conceptualization, X.Z. and X.H.; Data curation, T.X.; Formal analysis, X.Z. and T.X.; Methodology, X.Z.; Project administration, X.H.; Writing—original draft, X.Z.; Writing—review & editing, X.H.. All authors have read and agreed to the published version of the manuscript.

**Funding:** This research was funded by the National Natural Science Foundation of China, grant number 51576097, and Funding of Jiangsu Innovation Program for Graduate Education, grant number KYLX16_0357, and the Fundamental Research Funds for the Central Universities, the grant number 3082018NP2018421.

**Conflicts of Interest:** The authors declare no conflict of interest.

## Nomenclature

*Symbols*

| | |
|---|---|
| $W_f$ | Fuel flow |
| $\beta$ | Propeller pitch angle |
| $N_h$ | Core machine speed |
| $N_p$ | Propeller speed |
| $M$ | Shaft torque |
| $P_v$ | Propeller absorbed power |
| $C_p$ | Power coefficient of propeller |
| $J_p$ | Propeller advance ratio |
| $V_0$ | Axial velocity of propeller |
| $R_p$ | Radius of propeller |
| K | Dynamic coefficient |
| $P_e$ | Power turbine output power |
| $P_v$ | Propeller absorbed power |
| $J$ | Inertia moment |

*Definitions, acronyms and abbreviations*

| | |
|---|---|
| MGTE | Micro gas turbine engine |
| UCAV | Unmanned combat air vehicle |

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
