# Peer review of "Research on Modeling of a Micro Variable-Pitch Turboprop Engine Based on Rig Test Data"

_energies, doi:10.3390/en13071768_

Round 1

Reviewer 1 Report

The text of the article is made with violations of the editorial template.
A review of literary sources is very litle.

There is no sufficient substantiation of mathematical formulas. An explanation is needed.

Conclusion - add numerical values. back up with your research links and graphs. display further research and perspectives.

Author Response

Dear reviewer:

Thank you for your professional comments. We have studied your comments carefully and have made revision according to the comments. All of the suggestions are reflected in the revised paper or in this letter. Revisions are highlighted in the paper in red font.

  1. The whole article is updated based on a standard template.
  2. Some references from the last 5 years are added and elaborations about them are added in the introduction sections.
  3. Some elaborations and omitted steps about the formulas are added, as shown at line 162 and 163, and from line 283 to 288.
  4. More elaborations are added in the conclusion part, as well as the margin of error. Further research plans are added, from line 325 to 333. The results graph is shown in the part just above the conclusion, so it is not shown in this part.

Reviewer 2 Report

The article presents the modeling method based on rig data is proposed to establish a double-input double-output model for a micro turboprop engine. Generally, the research part is correctly planed and described. However, many elements need improvement as follows:

#1  Article formatting

The article is badly formatted. The issues that must be changed in accordance to Energies temple:

-> style of abstract

-> style of sections names

-> style of the main text (it cant be Calibri, it should be Palatino linotype)

-> the description of the figures should have a dot after number and text should start with capital letter and end with dot e.g. “Figure 1. Overall structure of the model.” The same situation with tables.

-> style of references list is not correct

-> references  - ” In the text, reference numbers should be placed in square brackets [ ], and placed before the punctuation”

Generally, please beck to temple and use the styles that are available in the document. In this form it’s unacceptable.

#2 Article type:

Please add the type of the paper in the first line. “Type of the Paper (Article, Review, Communication, etc.)“

#3 Correspondence and article information :

Please add as in Energies temp: “Correspondence: [email protected]; Tel.: (optional; include country code; if there are multiple corresponding authors, add author initials) +xx-xxxx-xxx-xxxx (F.L.)” and “Received: date; Accepted: date; Published: date”

#4 Keywords

Please add the keywords to the text

#5 Introduction section

-> Please add at the end of the introduction, the short description of each article section. E.g. “The article is organized as follows: section 2 describes … Section 3 … etc.”

-> Please add the actual state of the art from literature.  Actually, in described solutions (references 8-15) there is a lack of references from the last 5 years (2016-2020) only “old” articles are included.

#6 Units

Use space between value and unit e.g. “100daN” should be “100 daN”

#7 Figures

-> Please correct Fig. 1 - Propeller speed Np it’s not fully visible

-> Please correct the descriptions of Fig. 2 -> the letters should be bigger.

-> Fig. 10: the description “propeller absorbed …” should start with a capital letter

#8 Table

The quality of the table is unacceptance. Please check in Energies temple to look how the table may look like.

#9 Discussion

Please add the discussion section. The elements that should be included are like in Energies temple description:

“Authors should discuss the results and how they can be interpreted in perspective of previous studies and of the working hypotheses. The findings and their implications should be discussed in the broadest context possible. Future research directions may also be highlighted.”

#10 acknowledgement / foundning:

Please back to Energies temple and add correctly founding describtions.

#11 Authors contribution

Please add the author's contributions as in Energies temple.

#12 Declaration

Please add the conflict of interest declaration as in Energies temple.

# 13 References describtion

Please check all the references. There are not correctly described. Please back to Energies temple.

#14 Specific remarks:

  • Please delate the yellow highlight in text.
  • Line 117 The Fig. 2 and Fig. 3 should be introduced not Fig. 1 and Fig. 2
  • Lines 126-142. Please use a bullet list. In the continuous form, it’s unreadable. Please change and clarify it.
  • Line 169 Please write “Figures 4-9” not “Figure 4-9”
  • Line 176 as in line 169
  • Line 184 Please add a dot after sentence.
  • Line 222 Please add the number of the figure
  • Line 255 Please change the color of bracket.

Author Response

Dear Reviewer:

Thank you for your commands and I appreciate your carefulness very much. We have studied your comments carefully and have made revision according to the comments. All of the suggestions are reflected in the revised paper or in this letter. Revisions are highlighted in the paper in red font.

  1. The whole article is updated based on a standard template. The styles of the abstract, sections names, main text, the description of the figures and the references list are all updated.
  2. The type of the paper is article, and it is added in the first line.
  3. The correspondence information is added. The received data is added.
  4. The keywords are added at line 26.
  5. A short description of each article section is added at the end of introduction, from line 89 to 93. Some references from the last 5 years are added and elaborations about them are added in the introduction sections.
  6. The “100daN” is changed into “100 daN” at line 29.
  7. The arrows in Fig.1 are adjusted to make Np visible. Letters in previous Fig.2, now cited as Fig.2(a), are enlarged. The description “propeller absorbed …” in previous Fig.10, now cited as Fig.4, is changed into capital letter.
  8. The table is adjusted based on the temple.
  9. A discussion section is added. The future research directions are added in this section and the conclusion section.
  10. The founding part is updated according to the temple at line 360.
  11. Author’s contributions are added as in energies temple at line 356.
  12. Conflict of interest is added at line 363.
  13. The references are adjusted based on energies temple.
  14. The yellow highlight in the text is delated, and the revisions are highlighted as red font.

Line 116 which is line 117 in previous version, the Figure 1 and Figure 2 is changed to Figure 2 since the cited number is changed based on the temple.

Line 127-144, which was line 126-142, a bullet list is used. For more clear expression, this part is reorganized.

Line 175, which was line 169, it is changed into “Figure 3”’

Line 182, which was line 176, it is changed into “Figures 3(c)-(f)”.

Line 189, which was line 184, a dot is added.

Line 231, which was line 222, the figure number is added.

Line 265, which was line 255, the color of bracket is revised.

Reviewer 3 Report

There are some points that need revising:

  • More details regarding the filtering process of the data in Section 4 should be provided
  • The manuscript must be improved to comply with the journal template. More specifically, citing of references in the main body, the font of the Headings, etc.
  • The legends should not overlap and cover the curves in the figures, e.g. Fig. 4 - Fig. 7, and many more

Author Response

Dear reviewer:

Thank you for your commands and I appreciate your carefulness very much. We have studied your comments carefully and have made revision according to the comments. All of the suggestions are reflected in the revised paper or in this letter. Revisions are highlighted in the paper in red font.

  1. Some explanations about the fluctuations and elaborations about the filter are added in section 4. There is a mistake in the last version. The data displayed in this manuscript is processed by a mean filter, not inertial filter. Actually inertial filter with filter time 20ms is also ok in this situation.
  2. The whole article is updated based on a standard template now.
  3. All the data graphs are adjusted and no curve is covered by the legends now.

Reviewer 4 Report

Minor English improvement is required, for example instead of “with enough accurate” it is better “with enough accuracy”, also “Nether” is “neither”?

At line 94, the title of Fig 1 must be with capital letter. At line 117 I think that is not about Figure 1 and Figure 2 but it should be Fig 2 and 3.

In my opinion is a well-written, well-organized and well-illustrated paper. The conclusions are supported by the presented information and therefore the paper can be accepted for publishing.

Author Response

Dear reviewer:

Thank you for your commands and support very much. We have studied your comments carefully and have made revision according to the comments. All of the suggestions are reflected in the revised paper or in this letter. Revisions are highlighted in the paper in red font.

  1. At line 126, “accurate” is changed into accuracy. At line 183 “Nether” is changed into “neither”;
  2. Line 96, which was line 94, the title is changed into capital letter. Line 116 which is line 117 in previous version, the Figure 1 and Figure 2 is changed to Figure 2 since the cited number is changed based on the temple.

Round 2

Reviewer 1 Report

great result

Author Response

Dear reviewer: On behalf of my co-authors, I would like to express our thanks to you for your help and support. Some adjustment in format are made and some discussion of the present literature are added in this version.

Reviewer 2 Report

Dear Authors. I would like to congratulate You. Your paper has been improved. All elements from the previous review were included in text. However, I would like to say, that deeper discussion of the present literature, for future work, would be really desirable :)

The last element is that references are still not in accordance with the Energies temple, please change it before final publication.

Author Response

Dear reviewer:

Thank you for your commands and support :). 

The references are adjusted according to the Energies temple by using the software Endnote. Some discussions of the present literature are added.

Reviewer 3 Report

The authors have satisfactorily replied to my comments

Author Response

Dear reviewer:

Thank you for your commands and support. 

Some adjustments in format are made and some discussions of the present literature are added in this version.

This manuscript is a resubmission of an earlier submission. The following is a list of the peer review reports and author responses from that submission.

Round 1

Reviewer 1 Report

Existing literature on this subject needs to be expanded to justify novelty. There are very few references (11) and only three from the past 5 years. This is very surprising given the growth of this technology in micro gas turbine based power generation.

The order of sections is not logical. There is a theory section late on in the paper. The discussion is weak - there is very little discussion on the significance of the work. The 'two major conclusions' mentioned in the conclusions section are not elaborated.

Formatting is poor. Figures should be on separate lines. There are no dimensions presented on the schematic. Figure captions are placed very close to the text making them indistinguishable from the text. Table caption is on a separate page to the table. Some figures are too small.

The figures are not explained in much detail. Rather they are used as a means to evidence that data was gathered. They should instead complement the text descriptions of the outcomes of the experiment and modelling. There is also no explanation for the shape of the spline fit to the data points. What is the reason for the undulation in the line between data points for example in the fuel flow in Figure 6.

The 'dynamic process' arising from the torque mismatch needs to be elaborated - what is meant by this term, why does it happen, and what is the significance of this?

The level of English is poor. Use 'et al.' instead of 'etc.' when referring to authors. There are grammatical errors throughout, and many spelling mistakes. I began to correct them but it was taking too much time - this is not the job of a reviewer.

The presentation is messy - graphs are not aligned, fonts change, equation symbols are not always defined and there is no nomenclature. Some equations are unfinished - with parentheses missing e.g. Eq.11.

The method for calculating the error is not clear.

Reviewer 2 Report

This paper investigates a simplified modelling method for input output tests of a turboprop engine.

The experiments of Figures 4-9 are very conservative and unlikely to excite any significantly non-linear dynamics. What happens to the propeller response for larger variations and faster frequencies in the beta? Should at least provide some frequency responses to see if higher order dynamics are present. Where are the large fluctuations in the data? Should present raw data as well as filtered data. How identifiable are the parameters when there is so much noise? Should show the sensitivities of the identified parameters with respect to different filters from raw data up to heavily filtered data. What errors are their in the transient response prediction of the model by using the base-line correlation of Figure 11. Should show a comparison with using the hysteresis curve. How consistent is the hysteresis curve with respect to different operating points, e.g. changes in beta? Equation (15) does not look corret. Where is the damping? Why is there a Np’(t) term on the bottom line? Where does the 30 come from? Also, you’ve assumed the rotor acceleration is proportional to P_e(t)-P_v(t). Under what circumstances is this true? Do P_e(t) and P_v(t) have to be at steady state, or only slowly varying? Why use Euler’s method for Equations (10) and (16)? This is well known to be very slow computationally. There is no information given on what data is extrapolated based on the model and what part is identified. E.g. how well does the data extrapolate N_p and N_h for greater propeller speeds which are outside the bounds of the modelled data? How valid is Equation (1) during transient conditions? Should compare this simplified model to the more complex models in the literature. What is the control application? Should at least show via simulation why the proposed model would provide a better control. How do computational requirements for the method compare with other methods and if this modelling is done offline, why is computational speed important? There are many grammatical errors in the paper